# The Role of Autophagy in Osteoarthritic Cartilage

**DOI:** 10.3390/biom12101357

**Published:** 2022-09-23

**Authors:** Wei-Chun Kao, Jian-Chih Chen, Ping-Cheng Liu, Cheng-Chang Lu, Sung-Yen Lin, Shu-Chun Chuang, Shun-Cheng Wu, Ling-hua Chang, Mon-Juan Lee, Chung-Da Yang, Tien-Ching Lee, Ying-Chun Wang, Jhong-You Li, Chun-Wang Wei, Chung-Hwan Chen

**Affiliations:** 1Orthopaedic Research Center, College of Medicine, Kaohsiung Medical University, Kaohsiung 80708, Taiwan; 2Department of Orthopedics, Kaohsiung Medical University Hospital, Kaohsiung Medical University, Kaohsiung 80708, Taiwan; 3Regeneration Medicine and Cell Therapy Research Center, Kaohsiung Medical University, Kaohsiung 80708, Taiwan; 4Department of Orthopedics, College of Medicine, Kaohsiung Medical University, Kaohsiung 80708, Taiwan; 5Department of Medical Education and Research, Kaohsiung Veterans General Hospital, Kaohsiung 813414, Taiwan; 6Department of Orthopedics, Kaohsiung Municipal Siaogang Hospital, Kaohsiung 812, Taiwan; 7Department of Medical Science Industries, Chang Jung Christian University, Tainan 71101, Taiwan; 8Department of Bioscience Technology, Chang Jung Christian University, Tainan 71101, Taiwan; 9Graduate Institute of Animal Vaccine Technology, College of Veterinary Medicine, National Pingtung University of Science and Technology, Pingtung 912301, Taiwan; 10Department of Orthopedics, Kaohsiung Municipal Ta-Tung Hospital, Kaohsiung 80145, Taiwan; 11Department of Healthcare Administration and Medical Informatics, Kaohsiung Medical University, Kaohsiung 80708, Taiwan; 12Ph.D. Program in Biomedical Engineering, College of Medicine, Kaohsiung Medical University, Kaohsiung 80708, Taiwan; 13Institute of Medical Science and Technology, National Sun Yat-sen University, Kaohsiung 80420, Taiwan; 14Graduate Institute of Materials Engineering, College of Engineering, National Pingtung University of Science and Technology, Pingtung 912301, Taiwan

**Keywords:** AMP-activated protein kinase (AMPK), Autophagy, circular RNAs, microRNAs, long non-coding RNAs, non-coding RNAs, Osteoarthritis (OA)

## Abstract

Osteoarthritis (OA) is one of the most common diseases leading to physical disability, with age being the main risk factor, and degeneration of articular cartilage is the main focus for the pathogenesis of OA. Autophagy is a crucial intracellular homeostasis system recycling flawed macromolecules and cellular organelles to sustain the metabolism of cells. Growing evidences have revealed that autophagy is chondroprotective by regulating apoptosis and repairing the function of damaged chondrocytes. Then, OA is related to autophagy depending on different stages and models. In this review, we discuss the character of autophagy in OA and the process of the autophagy pathway, which can be modulated by some drugs, key molecules and non-coding RNAs (microRNAs, long non-coding RNAs and circular RNAs). More in-depth investigations of autophagy are needed to find therapeutic targets or diagnostic biomarkers through in vitro and in vivo situations, making autophagy a more effective way for OA treatment in the future. The aim of this review is to introduce the concept of autophagy and make readers realize its impact on OA. The database we searched in is PubMed and we used the keywords listed below to find appropriate article resources.

## 1. Introduction

### 1.1. Osteoarthritis

Osteoarthritis (OA) is a very common degenerative joint disease influencing many people by causing pain, stiffness and disability, especially for the elderly [1]. About one-third of people aged more than 65 are in pain with OA [2]; it costs a great deal of socioeconomic burden owing to the increasing prevalence caused by complicated and multifactorial situation including occupation, food habits, hormones, and heredity [3].

The global prevalence of knee OA is around 3.8% [4]. There is a report revealing that OA is the second-most cause, around 7.1%, of musculoskeletal diseases’ burden, and the burden of OA considerably raised 63.1% from 1990 to 2007 [5]. Besides, OA is one of the most challenging diseases to remedy since there are no blood vessels and nerve supplies involved in mature articular cartilage. The regeneration ability is weak and usually, fibrous cartilage is formed [6].

Symptomatic OA requires medical intervention. Nevertheless, the therapeutic choice is still limited and no disease-modifying drug is available. The degenerative process of osteoarthritic cartilage in aging patients is not able to be completely delayed [7,8]. Furthermore, surgery like total knee arthroplasty is risk and costly. Hence, an alternative approach has been revealed. In addition to the application in cancer, autophagy may be a surrogate for OA, which numerous pieces of research are focusing on [9].

### 1.2. Pathogenesis of Osteoarthritis

OA is recognized as a whole joint disease involved in cartilage, infrapatellar fat pad, subchondral bone, ligaments, meniscus and synovial membrane, which eventually causes joint failure [10]. There are several risk factors to increase the peril of developing OA. Diabetes elevating inflammatory response, causing atherosclerosis to affect blood flow of subchondral bone is thought to be contributory to OA [11]. Post-menopausal women have raised their risk of OA due to lower estrogen level and lack of ability to protect cartilage from oxidative stress [12]. Obesity which overloads the weight-bearing system of our body is also a crucial part of inducing OA [10]. Besides, about 12% of OA can be caused by trauma immediately or chronically which forms alterations in biomechanics whether it is structural damage or joint instability [13,14].

However, the pathogenesis of OA is mostly about aging. The imbalance of catabolism and anabolism in the cartilage matrix is related to aging because the more vulnerable joint cannot afford damage from outside [15]. Hemostasis could not be maintained by increasing catabolic along with decreasing the anabolic pathway of chondrocytes which leads to progressive degradation of cartilage [16]. In histology, the main pathogenesis of OA is the terminal differentiation of chondrocytes and the subsequent apoptosis. Chondrocyte apoptosis is also an important issue in OA. Chondrocyte apoptosis is associated with OA in the feature of cartilage and extracellular matrix devastation, which was proved by electron microscopy to show a few characterized cellular morphology in the sample of osteoarthritic cartilage, such as loss of nuclear volume, apoptotic bodies, cell-surface blebs and even the alteration of nuclear shape [17,18].

We can outline OA as early and late stages [1]. Not only growth of the cellular size but also congregate chondrocytes are found in the early stage, but the compensation is not enough for the cell matrix as the amount of glycosaminoglycan (GAG) starts to diminish. However, in the late stage, situations become more severe along with chondrocyte death, hypo-cellularity and lacunar emptying [1,19].

Besides, the situation of autophagy (macro-autophagy) and apoptosis in cartilage may change depending on the stage of OA. Early stage of OA demonstrated both elevated markers of autophagy and apoptosis in the superficial zone, indicating that autophagy is an adaptive response to apoptotic cell death, while in late stage of OA, in the deep zone, more apoptosis with the absence of autophagy represented the chondrocyte calcification [20]. Then, there is a type of chondrocyte death called chondroptosis, also related to autophagy proved by the existence of autophagic vacuoles [21]. Almonte-Becerill proposed that this cell death is like a mixture of autophagy and classical apoptosis [20]; it is the concept appearing in the late stage of OA as both apoptosis and autophagy are activated at this time in chondrocytes to show a different way of cellular demise [20].

## 2. Autophagy

### 2.1. Definition, Classification and Mechanism

#### 2.1.1. Definition

Cellular components are constantly recycled and remodeled in humans. There are two major systems of degradation in eukaryotic cells; one is proteasomal degradation which selectively identifies ubiquitinated substrates, and the other is lysosomal degradation [22]. In lysosomal degradation, protein, organelle, cytosolic components and even extracellular substances would be transferred to lysosomes through autophagic activity wherever they are in animals, plants or yeast cells [22,23]. Thought to be nonselective in general. However, autophagy has been noticed to selectively remove possibly detrimental or unnecessary cytoplasmic material lately. Aggrephagy means eliminating accumulated proteins; meanwhile, mitophagy is the way to take off mitochondria (damaged organelle) [24,25].

We can find the different functions of autophagy in previous studies. Hunger response is the process that promotes autophagy and decomposes large molecules to intermediate metabolites used in anabolism or katabolism, for example, the amino acid deficiency in the liver [26]. When it comes to regulating cell activity, autophagy is responsible for degrading normal proteins and recombining cells during aging, differentiation and metamorphosis among animals [27,28].

#### 2.1.2. Classification

Autophagy could be categorized as three types by physiological functions: chaperone-mediated autophagy, microautophagy, and macro-autophagy [29,30], and macro-autophagy is mostly discussed in the field of physiology, cytology and pathology. Besides, macro-autophagy utilized at the beginning to take away unemployed proteins and damaged organelles is the main pathway of mammalian autophagy. Microautophagy functions as maintaining organelle size and membrane homeostasis by engulfing small cytoplasmic components [31], while chaperone-mediated autophagy degrades proteins in a more selective way, such as pentapeptide recognized by hsc70 (heat-shock cognate protein of 70 kDa) in the amino acid sequence [32].

#### 2.1.3. Mechanism

There are a few key steps involved in macro-autophagy. First is the elongation of phagophores to produce autophagosome (Figure 1). The formation of autophagosomes, cytosolic double-membrane vesicles segregating part of the cytoplasm is crucial in macro-autophagy [33,34]. There are mainly two membrane sources autophagosomes are from. In yeast, the membrane is from different organelles while another one is the omega-shaped structure from omegasome, the endoplasmic reticulum subdomain enriched in phosphatidylinositol-3-phosphate (PtdIns3P) [35]. The origin of the autophagosome is a phagosome that also needs the help of autophagy-related gene (Atg) proteins joining into a phagophore assembly site (PAS) [36].

The process from phagophore to autophagosome depends on the activity of two ubiquitin-like conjugation systems (Figure 1), the Atg5–Atg12 system and the microtubule-associated protein-1 lightchain 3 (LC3, Atg8) system [37]; the ubiquitin chain, also known as the “eat-me” signal label, is identified by the autophagic receptor. In Atg5–Atg12 system, the ubiquitin-like protein of Atg12 attaches to Atg5 with a covalent bond, and combines with Atg16L1 to form the group: Atg12–Atg5–Atg16L1, which is indispensable for the elongation of the pre-autophagosomal membrane [38,39].

The other axis is microtubule-associated protein-1 lightchain 3 system related to lightchain 3 (LC3), also known as Atg8. The precursor form, LC3B, is cleaved by protease Atg4B at its C-term, and turns into the cytosolic free-form (LC3-I). Then, the alteration from LC3-I to LC3-II (phosphatidylethanolamine-conjugated) is the beginning of autophagy [40,41,42,43].

Second, fusion: autophagosomes fuse with endosomes and lysosomes to become an autolysosome which is controlled by the GABARAP subfamily (Figure 1). The GABARAP subfamily helps recruit Pleckstrin homology domain-containing protein family member 1 (PLEKHM1), which acts on the homotypic fusion and protein sorting (HOPS) complex and its contained LIR (LC3-interacting region), stimulating the fusion of endosomes and autophagosomes with lysosomes as a consequence [44]. By the way, LC3-Ⅱcontinues to target to the elongating autophagosome membrane until its fusion with lysosomes, and changes into autolysosomes breaking down the materials contained within [40].

There is evidence that AMP-activated protein kinase (AMPK) also promotes autophagosome and lysosomal fusion [45]. Jang (2018) and colleagues mentioned using trehalose or compound C to treat cells knocked out of AMPKα1 mostly inducing the stimulation of autophagosomes instead of autolysosomes, suggesting that the fusion necessitates AMPKα1.

After the integration, there are degradation and recycling; encapsulated large molecules divided into lipids, amino acids, energy and nucleotides are prepared for subsequent cellular renovation, homeostasis or response to the cell stress [46,47].

### 2.2. Role of Autophagy in Osteoarthritis

OA is linked to chondrocyte apoptosis. Autophagy is thought to be in balance with apoptosis when there is increased chondrocyte apoptosis occurring with lower expression of autophagy regulators in OA joints of human beings and rodents [48]. Autophagy seems to be a protective mechanism for stabilized microenvironment in articular cartilage and less autophagic activity is observed in aging or osteoarthritic joints [49].

Autophagy in OA can be seen in the aspect of autophagy genes. Initially, autophagy mRNA of Beclin-1 (BECN1) and LC3 is increased which is considered a compensation against cellular stress. In mild OA, difference start to happen in the superficial zone of articular cartilage: protein expression levels of BECN1, LC3 and ULK1(an autophagy-initiating kinase) become lower, thought to be defective autophagy. Then, there is an all-layered (including middle and deep zones) autophagic genetic change not only reduced ULK1 (Unc-51-like kinase 1), LC3 and BECN1 but also fewer ATG genes (ATG3, ATG5, ATG12) in advanced OA [48,50,51,52]; furthermore, chondrocyte apoptosis arise as markers like PARP and p85 elevating.

In mouse OA, loss of proteoglycan, one of the principal contents to the extracellular matrix, was found to be associated with reduction of the autophagic markers [48,53]. Some knockout research also proved the connection between OA and autophagy with ablation of ATG5, ATG7 [54,55].

## 3. Signaling Pathway Related to Autophagy in Osteoarthritis

### 3.1. AMPK-SIRT1-FOXO3A

AMPK is a heterotrimeric complex comprising catalytic α subunit and regulatory β-, γ-subunits [56]; it is also an important energy sensor balancing energy in control of protein, glucose or lipid metabolism. In addition, AMPK promotes autophagy and has its own role in OA (chondroprotection) thought to be related to redox regulation and mitochondrial function (mitophagy) [57,58,59]. AMPK can stimulate mitophagy for the removal of damaged mitochondria which is blocked by mTORC1 [60,61].

There are reports showing that AMPK could phosphorylate and activate the downstream targets such as sirtuin1 (SIRT1) and Forkhead box class O 3a (FOXO3a) (Figure 1), then restraining the injury and gene expression of inflammation [59], while FOXK1 and FOXK2 (members of the FOX family) phosphorylated by mTOR in high nutrient condition would work against FOXO3 to suppress autophagic gene activity [62].

SIRT1 and FOXO3a are mentioned to promote autophagy-related protein expression (BECN1, LC3), deacetylate autophagic proteins (ATG5, ATG7, and ATG8) and help assembly of autophagosome [63]. For example, hydroxytyrosol, which is found in olive oil and shows antioxidant properties, encourages the transfer of SIRT 1 to the nucleus. SIRT 1 into the nucleus is speculated to catalyze deacylation of LC3-II, and LC3-II is back to cytosol for autophagy initiation. Thus, hydroxytyrosol is thought to be beneficial when chondrocytes face oxidative stress [64,65]. In the SIRT family, SIRT 1 is most relevant to OA through autophagy, mitochondrial biogenesis and other response. 17β-Estradiol is also proved to promote chondrocyte protection by mitophagy through the AMPK/SIRT1/mTOR pathway [66].

Promotion of autophagic gene expression by FOXOs has occurred in neurons, muscle and even hematopoietic stem cells [67,68], and there is FOXO3 which elicit autophagy in chondrocytes of mouse and human [69]. Glucosamine, which is a common dietary supplement, also promotes autophagy in cartilage through activating FOXO3 (dephosphorylation); in contrast, glucocorticoids can stimulate FOXO3 and activate autophagy through a higher level of ROS [70,71].

### 3.2. AMPK-ULK1

ULK1, homologue of yeast ATG1, is a serine/threonine kinase in mammals; Among five members of ULK1 homologues, there are ULK1 and ULK2 participating in autophagy and ULK1 is thought to be an inducer of the process [72]. ULK activates BECN1 by phosphorylation and combines to form a protein complex, relocating to site of the phagophore to initiate the composition of the autophagosome. Besides, there is interaction between AMPK and Ulk1 when the level of glucose is changed (Figure 1). Active AMPK promote Ulk1 via phosphorylation of serine 317 and serine 777, strengthening autophagy under nutrient deprivation [73]. On the other hand, AMPK can also inhibit mTOR by phosphorylation through the intermediate regulator TSC2 [74]. Besides, there is report indicating the association between ULK1 mutants and the accumulation of defective mitochondria which implies the effect of ULK1 on mitophagy, a core event in OA as well [75].

Some instances are described through the AMPK/ULK1 axis. Ansari et al. observed that butein, an organic compound belongs to the polyphenol family (found to be useful in OA management), triggers autophagic flux through phosphorylation of AMPKα threonine-172 and ULK1 serine-317, and suppresses mTOR in OA-like chondrocytes [76]. Then, there is one research about PLCγ1 (Phospholipase C gamma 1) speaking about the indirect reaction of AMPK and ULK1 with the utilization of AMPK activator metformin decreasing the level of p-PLCγ1 and blocking mTOR/ULK1 axis, eventually encouraging the synthesis of cartilage extracellular matrix [77].

## 4. Non-Coding RNA and Autophagy in Osteoarthritis

### 4.1. Introduction of Non-Coding RNA

Nucleic acids were discovered in 1869. Then, after quite a long time, more than a hundred years, regulatory roles of non-coding transcripts start to be identified and discussed [78]. The very first regulatory non-coding RNA (ncRNA) was found from E. coli discovered in 1984 by Masayuki Inouye which was a bacterial small RNA (sRNA) [79].

98% of the genome is known to be transcribed, but most of the transcripts do not encode proteins and they are called ncRNAs [80]. Lots of studies have revealed ncRNAs to be associated with regulating autophagy in various diseases, for example, diabetes, cancer, myocardial infarction and so on [81].

There are many types of ncRNA, regulatory ncRNAs such as microRNA (one of the small ncRNAs), long non-coding RNA (lncRNA) and circular RNA (circRNA) which belongs to the lncRNAs; all these three types of ncRNAs are commonly involved in the process of autophagy in OA, and genes can be modulated at different stages including epigenetic, transcriptional, and post-transcriptional levels [82,83]. In addition, there is a complicated chain called the competing endogenous RNA (ceRNA) network adopting microarrays to analyze and display reactions between ncRNAs (microRNA, lncRNA and circRNA) and mRNA which can help us to detect the detailed molecular mechanism and uncover more OA-related genes [84,85].

### 4.2. MicroRNAs and Autophagy in Osteoarthritis

MicroRNA (miRNA) is a posttranscriptional modulator that has been involved in cartilage homeostasis and regulated biological processes of chondrocyte autophagy by targeting genes; especially for its widespread target genes, miRNA have an impact on lots of diseases, such as cancer, inflammatory bowel disease, myocardial infarction and so on [81,86,87,88,89].

Conserved sequences have been the key point making miRNAs an interesting topic for researchers. The details of miRNA composition originate from primary RNA (pri-miRNA) which is cleaved into a hairpin around 70 nucleotides and then become pre-miRNA. Pre-miRNA is transported by Exportin-5 from the cell nucleus to cytoplasm; after that, with an RNase III called Dicer changing the structure into a double-stranded miRNA about 22 nucleotides left. Consequently, a biological complex named miRNA-induced silencing complex (mRISC) was formed by the composition of mature miRNA and proteins of the argonaute (AGO) family [90,91,92,93,94].

This complex identifies certain genes and binds to the 3′-UTR of targeted mRNAs. Furthermore, the extent of complementary base pairing decides the intensity of regulation. Inhibition of mRNA translation is elicited by partial complementary base pairing; on the other hand, cleavage and degradation of mRNA occur when there is perfect complementarity [94,95].

MiRNAs have been found to modulate SIRT1 and influence OA progression. By decreasing SIRT1 expression, miR-9, miR-34a, or miR-449a possibly cause cartilage destruction; hence, inhibiting these miRNAs may be a therapeutic target for OA [96,97,98]. Furthermore, D’Adamo (2016) and their colleagues mentioned miR-155 is quite crucial in autophagy in OA, thought to suppress the autophagic activity through targeting ULK1, FOXO3 (a key transcription factor of ATG genes) and ATG14 [99].

Several miRNAs have also been engaged in OA with other mechanisms. MiR-27a enhanced autophagy in IL-1β-treated chondrocytes via downregulating PI3K [100]. MiR-107 was unveiled with the promotion of autophagy by targeting TRAF3; miR-411 with the target of HIF-1alpha mRNA showed the ability to strengthen autophagy in OA models [101,102]. Contrarily, miR-375 was elevated in the samples from osteoarthritic cartilage possibly through inhibition of ATG2B and restraining autophagy along with more cellular stress [103]. MiR-378 aggravated OA via suppressing ATG2A and the subsequent autophagic activity [104].

In addition to the miRNAs mentioned above, still lots of miRNAs are being investigated. Many possible candidates emerge due to their differential expressions in OA cartilage, and involvement in autophagic activity in non-cartilage tissues [105]. Other reasons may be their high cartilage specificity or sharing the same target with the miRNA which has been confirmed to mediate autophagy in chondrocytes. Nevertheless, their precise pathway or mechanism, and those associated proteins, genes or cytokines must be elucidated.

### 4.3. lncRNAs and Autophagy in Osteoarthritis

LncRNAs were investigated in the late 1980s with more than 200 nucleotides. H19 was the first eukaryote lncRNA gene localized to mouse chromosome 7 which has a key role in embryonic development [106], and changed our concept for the biological relevance of lncRNAs. Then, in the 2000s, a worldwide effort to lncRNAs began.

LncRNAs are produced by RNA polymerase II and III, capable of modulating protein-coding genes in positive and negative directions [107]. The regulatory mechanism of long noncoding RNAs is intricate and related to chromatin, histone, mRNA and so on [108,109]. There are also connections between lncRNAs and miRNAs; for instance, they can combine to form regulatory networks which indirectly affect gene regulation of miRNAs; on the other hand, lncRNAs may serve as miRNA precursors transforming into miRNAs through RNases [110,111].

A few lncRNAs have been noticed to participate in OA through autophagy. Song (2014) et al. led the way to find GAS5 (growth arrest-specific 5) which already had a role in apoptosis [112,113]. GAS5 suppressed miR-21 which controlled the expression of type II collagen, aggrecan and MMP13, and decreased autophagic activity in osteoarthritic chondrocytes. Ji (2021) et al. furthermore mentioned the level of GAS5 in the late stage of OA is superior to the one in the early stage, and silencing lncRNA GAS5 help the autophagy of chondrocytes. The reason may be the binding ability of GAS5 to miR-144 which modulates mTOR expression [114].

Another lncRNA being discussed is HOTAIR (HOX transcript antisense RNA); it is known to enhance proliferation, invasion and metastasis in preclinical studies about cancer [115,116], and highly expressed in cancer, such as acute myeloid leukemia [117]. Moreover, HOTAIR is relevant to autophagy and OA, thought to be associated with chondrocyte apoptosis in higher level of caspase 3 and less Bcl-2 expression. The mechanism of autophagy inhibition by HOTAIR is mediated by sponging miR-130a-3p which also suppress cell growth [118]. Later, Chinese medicine used to make knee OA therapy is found to protect the joint via chondrocyte autophagy in an animal study. Cangxitongbi (CXTB) promote autophagy, reduce HOTAIR expression, and ameliorate the cartilage structure [119]. Other lncRNAs regulating autophagy in OA are listed in Table 1 and Figure 2.

### 4.4. circRNAs and Autophagy in Osteoarthritis

CircRNA, which is not influenced by RNA exonuclease, was found to be a mediator of autophagy usually through targeting miRNA. Precisely, it performs as a miRNA sponge owing to abundant binding sites with higher stability and disease specificity, modulating those miRNAs through the site called MRE (miRNA response element) [128,129,130,131].

There is circRNA exacerbating tumor progression of cancer via enhancing the autophagic level [132]. CircRNAs have also been involved in OA microenvironment for control of autophagy to perturb the situation of inflammation [133,134]. Many more surveys about circRNAs in OA have emerged since 2015 [129], described below (Table 2) and showed in Figure 2.

#### 4.4.1. hsa_circ_0005567 (hsa_circ_0005567/miR-495/ATG14 Axis)

Zhang et al. found circ_0005567 sponged miR-495 to reduce ATG14 expression. Autophagy-related markers, LC3 and BECN1, and the ratio of LC3-II/LC3-I was upregulated by circ_0005567. Then, 3-methyladenine (3-MA), an autophagy inhibitor, overturned the promotion of autophagy mediated by circ_0005567 expression. In conclusion, the overexpression of circ_0005567 improved autophagy deficiency in OA chondrocytes and debilitated IL-1β-mediated chondrocyte apoptosis [135].

#### 4.4.2. ciRS-7 (ciRS-7/miR-7 Axis) 

In the research of Zhou (2020), they first found the level of ciRS-7 and miR-7 was not normal in OA. Then, in vitro study revealed autophagy inhibition and cartilage degradation mediated by IL-1β with down-regulation of ciRS-7 and upregulation of miR-7, which was thought to be an axis involved in the reaction by PI3K/AKT/mTOR activation. Furthermore, miR-7 mimics deteriorated destruction of cartilage in the animal model [136].

#### 4.4.3. circPan3 (circPan3/miR-667-5p Axis)

CircPan3 has its role in autophagy via targeting miR-667-5p; they show contrary validity towards chondrocytes which was proved in rat models. Lower expression of circPan3 was showed in the OA condition, and Zeng and colleagues demonstrated that circPan3 enhanced BECN1, LC3-II and Col2a1 expression and decreased the level of ADAMTS-5 and MMP13 [137].

#### 4.4.4. hsa_circ_0037658 (-) 

The link between hsa_circ_0037658 and autophagy was noticed in chondrocyte cell line, CHON-001 [138]. There is more hsa_circ_0037658 appearance in OA [139], and hsa_circ_0037658 small hairpin RNAs (shRNA) have the ability to decelerate apoptosis rate, overturn the increased MMP13 and decreased aggrecan, type II collagen-induced by IL-1β. Additionally, when hsa_circ_0037658 is knocked down, there is induction of autophagy: the same as hsa_circ_0037658 shRNAs reverse the reduction of LC3, ATG5 and BECN1.

#### 4.4.5. circRHOT1 (-) (circRHOT1/miR-142-5p/CCND1 Axis)

In the condition of OA, there is activated Cyclin D1 (CCND1), circRHOT1 and less miR-142-5p expression. miR-142-5p is negatively associated with CCND1 and circRHOT1 since circRHOT1 acted as miR-142-5p sponge to promote CCND1 in chondrocytes. Then, when circRHOT1 was knocked down, cartilage degeneration in OA model mitigated with raising level of aggrecan and type II collagen; even the autophagic marker: BECN1 and LC3 were elevated [140].

#### 4.4.6. circMELK (-) (circMELK/miR-497-5p/MYD88/NF-κB Axis)

In the study of Zhang (2022), it was constructed by using in vitro model to test circMELK, one of the most upregulated circRNAs in post-menopausal osteoporosis patients, for its effect on chondrocytes and see whether it will be a target for OA therapy. Autophagy was evaluated by LC3 and BECN1 expression. In sum, circMELK as a sponge of miR-497-5p modulated the action of the MYD88/NF-κB pathway, enhancing apoptosis and restraining autophagy of OA chondrocytes. However, the authors also mentioned the limitations of lacking enough sample size and in vivo experiments [141].

**Table 2 biomolecules-12-01357-t002:** CircRNA regulation of autophagy in OA.

First Author, Year	Experimental Design and Treatments	Results
Zhang, 2020 [135]	Model: human primary chondrocytes were added with recombinant human IL-1β (10 ng/mL) to stimulate degeneration for 24 h. 3-MA was used to pretreat chondrocytes. Treatments: Transfection with circ_0005567 overexpression vector	hsa_circ_0005567 Target/signaling pathway: miR-495/ATG14; LC3 ↑, BECN1 ↑, LC3-II/LC3-I ↑ Effects on chondrocytes: Promoting autophagy and inhibiting chondrocyte apoptosis
Sui, 2020 [138]	Model: In vitro-CHON-001 cells treated with IL-1β for 24 h; In vivo-15 Wistar rats (12-week-old) were randomly divided into three groups: control (n = 5), OA (n = 5) and OA + hsa_circ_0037658 shRNA1 (n = 5). Treatments: 24 h after surgery, rats in OA + hsa_circ_0037658 shRNA1 group were injected with lenti-hsa_circ_0037658 shRNA1 through the joint cavity twice a week.	hsa_circ_0037658 Target/signaling pathway: LC3 ↓, ATG5 ↓, BECN1 ↓, p62 ↑, AIF ↑ Effects on chondrocytes: Inhibiting autophagy; Knockdown of hsa_circ_0037658 alleviated the symptom of OA in vivo; Hsa_circ_0037658 shRNA reversed IL-1β-induced cell growth inhibition via inducing cell autophagy.
Zhou, 2020 [136]	Model: rats with DMM; C28/I2 chondrocytes stimulated with IL-1β for 24 h; OA articular cartilage samples and healthy cartilage samples from trauma patients without OA were collected. Treatments: The mixture of circRNA and miRNA was added to the cells for a 24 h transfection.	ciRS-7 Target/signaling pathway: miR-7; caspase 9 ↓, Bax ↓, Bcl-2 ↑ Effects on chondrocytes: Promoting autophagy and ameliorating cartilage degradation
Zeng, 2021 [137]	Model: IL-1β-induced rat chondrocytes and cartilage tissues of OA rats Treatments: Cell transfection	circPan3 Target/signaling pathway: miR-667-5p; ↑ BECN1, LC3-II, Col2a1, Acan; ↓ ADAMTS-5, MMP13 Effects on chondrocytes: Promoting autophagy
Man, 2022 [140]	Model: clinical OA samples, OA rats, chondrocytes Treatments: circRHOT1 shRNA	circRHOT1 (−) Target/signaling pathway: miR-142-5p/CCND1; ↓ BECN1, LC3, aggrecan and collagen II; ↑ COMP, CTX-II Effects on chondrocytes: Inhibiting autophagy and promoting chondrocyte proliferation, Knockdown of circRHOT1 could repress cartilage degeneration.
Zhang, 2022 [141]	Model: OA human cartilage tissue Treatments: sh-circMELK, cell transfection	circMELK (−) Target/signaling pathway: miR-497-5p/MYD88/NF-κB Effects on chondrocytes: Inhibiting autophagy and promoting chondrocyte apoptosis in OA

The relationship between ncRNA and its character for the autophagic activity in OA can be so intricate and need to be elucidated, yet it means the likelihood of keeping broadening the search for OA-related ncRNAs. Hopefully, they can serve as more comprehensive therapeutic targets or diagnostic biomarkers.

## 5. Prospective of Treatment via Autophagy in Osteoarthritis

There are wide variety of therapeutic options via autophagy in OA or for chondroprotection whether it is a hormone, antineoplastic agents and even supplements considered a novel strategy, for example, glucosamine could inhibit the mTOR pathway and trigger autophagy (Figure 1), thus helping to maintain cell homeostasis [142]. There is research showing that glucosamine has functions of anabolism and anti-inflammation for chondrocytes by controlling the molecular pathway of autophagy which is useful in animal study [70]. Growth of LC3 turnover and increment of LC3-II enhance the autophagic activity. Glucosamine also modulates cartilage cell apoptosis or survival, depending on time. Activation of autophagy is shown in short-term exposure of glucosamine; meanwhile, inhibition of autophagy is observed in the long period usage of glucosamine [143].

When it comes to parathyroid hormone (PTH), Chen and his partners found that PTH-(1-34) could improve the development of OA in an animal model by autophagy (Figure 1). In virtue of diminishing mTOR, reinforcing BECN1 and LC3, chondrocyte apoptosis may be weakened [144]. Vitamin D, a necessity for bone health, was also proved to alleviate inflammatory events of OA through AMPK-mTOR pathway [145]. Besides, the LC3-II: I ratio and level of BECN1 arose by administration of vitamin D, while its therapeutic effect was overturned by 3-MA, an autophagy inhibitor. Furthermore, Kong mentioned that vitamin D could be crucial in autophagosome aggregating in OA condition [145].

17β-Estradiol as the major female sex hormone have been found to take part in chondrocyte autophagy through ERK signaling pathway [146] (Figure 1). Ge et al. (2019) noticed a decreasing level of LC3 in the cartilage of the mice accepting ovariectomy. More LC3 conversion was observed and the inhibitory effect of compound C (AMPK inhibitor) would be offset when 17β-estradiol was used. Fenofibrate (FN), a drug for dyslipidemia, also shows the possibility for OA therapy. Lower expression of PPARα was found in OA mice and knees of OA patients. As a PPARα agonist, FN strengthened autophagic activity, preventing cartilage from degradation, and there is a retrospective study revealing the effect of fibrate to clinically ameliorating OA situation [147].

Rapamycin, also known as Sirolimus, is a drug with numerous effects which is often used in renal allograft rejection; its derivatives were approved for the treatment of advanced renal cancer carcinoma [148]. In addition, rapamycin showed its possibility in OA therapy by autophagy through lessening mTOR, stimulating LC3 and finally alleviating cartilage destruction. Local intra-articular administration of rapamycin in rodents seems to be effective [149], and decreased MMP13 and Col10a1 were identified after rapamycin treatment. Nowadays, multiple techniques combined to test therapeutic potential become more common. Xue et al. created a drug delivery system for cartilage exhibiting that via laser irradiation, rapamycin released from mesoporous polydopamine (MPDA) core could help chondrocyte preservation through activating autophagy [150]; this nanoplatform not only presents positive outcomes in the animal study but also implies plenty of opportunities for the progress of autophagy-related treatment.

## 6. Conclusions

OA is a public health issue with multiple mechanisms (inflammation, apoptosis, proliferation…) involved in articular cartilage. Autophagy is a characteristic process controlling the balance of the intracellular microenvironment, without an exception in chondrocytes. Then, this topic attracts thoroughly hard work and various aspects of comprehension which forms a molecular network added in many more genes and therapeutic agents; nonetheless, autophagy is not the only factor to decide the progress of OA. Crosstalk effects and multitargeting relationships should be taken into consideration. Comprehensive research of osteoarthritic cartilage function and successive clinical trials are necessities. Hopefully, there will be a disease-modifying strategy combined with different new medical techniques to be verified and treat OA fundamentally or those findings extend their utilized message into fields of inflammatory, degenerative diseases.

## Figures and Tables

**Figure 1 biomolecules-12-01357-f001:**
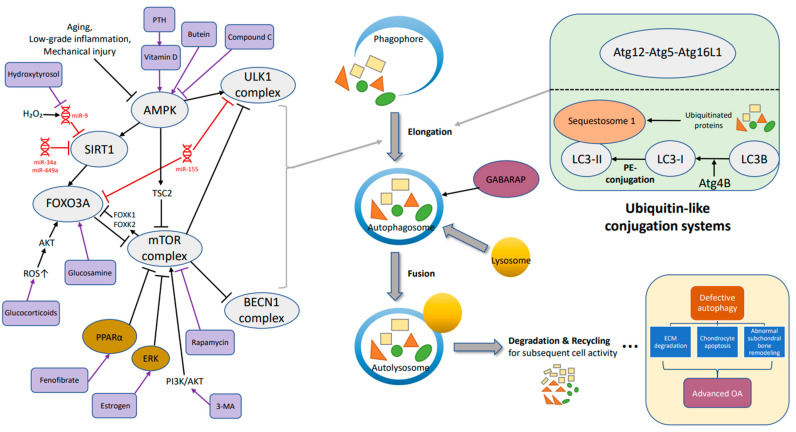
The mechanism of autophagy (key molecules and pathways) in osteoarthritis includes the following main steps: elongation, fusion, degradation and recycling. Defective autophagy could lead to extracellular matrix degradation, chondrocyte apoptosis and abnormal subchondral bone remodeling, resulting in osteoarthritis. →, promote; ┤, inhibit.

**Figure 2 biomolecules-12-01357-f002:**
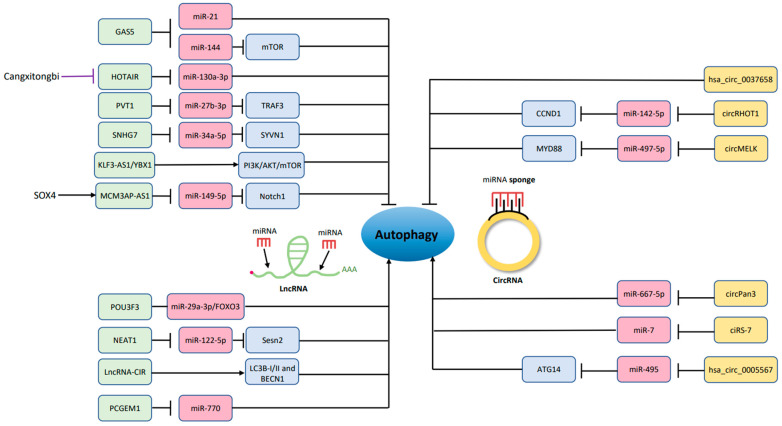
LncRNAs and circRNAs that promote or inhibit chondrocyte autophagy. The green color blocks (lncRNAs) and the yellow color blocks (circRNAs) modulate the pink color blocks (miRNAs) and the purple color blocks (other regulatory molecules) which result in altering the autophagic activity of chondrocytes. →, promote; ┤, inhibit.

**Table 1 biomolecules-12-01357-t001:** LncRNA regulation of autophagy in OA.

First Author, Year	Experimental Design	Results
Song, 2014 [112]	Model: articular chondrocytes isolated from relatively normal (Non-OA) and clear OA regions (OA) of cartilage in total knee replacement (TKR) patients; biopsied normal cartilage	GAS5 Target/signaling pathway: miR-21; LC3B; MMP2, MMP3, MMP9, MMP13, and ADAMTS-4 Effects on chondrocytes: Inhibiting autophagy and inducing apoptosis
Kang, 2015 [120]	Model: osteoarthritic human synoviocytes	PCGEM1 Target/signaling pathway: miR-770 Effects on chondrocytes: Inducing autophagy, inhibiting apoptosis and stimulating the proliferation of human synoviocytes
Wang, 2018 [121]	Model: rat/human; 8 patients undergoing total hip arthroplasty (THA), patients undergoing periacetabular osteotomy (PAO, n = 8)	LncRNA-CIR Target/signaling pathway: BECN1, LC3B-I/II; miR-130a/BIM; miR-27b/MMP13 Effects on chondrocytes: Inducing autophagy and apoptosis; CIR by regulating autophagy could promote articular cartilage degeneration in OA.
He, 2019 [118]	Model: knee OA-cartilage tissues located at the femoral condyles or tibial plateaus of OA resection regions compared with control regions in patients with normal non-weight-bearing area femoral condyle articular cartilage	HOTAIR Target/signaling pathway: miR-130a-3p; LC3; ↑ Bax, caspase 3; ↓ Survivin, Bcl-2 Effects on chondrocytes: Repressing autophagy, cell growth of chondrocytes, increasing apoptosis rates and reducing the cell viability
Lu, 2019 [122]	Model: cartilage tissues from 25 OA patients and normal controls, human transformed chondrocytes C28/I2 stimulated by IL-1β	PVT1 Target/signaling pathway: miR-27b-3p/TRAF3 axis; LC3B-II, BECN1; TNF-α Effects on chondrocytes: Silence of PVT1 promoted autophagy and cell viability but suppressed apoptosis and inflammatory response in IL-1β-treated C28/I2 cells.
Tian, 2020 [123]	Model: OA cartilage tissues from 15 OA patients, normal cartilage tissues from 10 patients; IL-1β induced OA chondrocytes	SNHG7 (−) Target/signaling pathway: miR-34a-5p/SYVN1; BECN1, MAP1LC3B Effects on chondrocytes: Upregulation of SNHG7 by sponging miR-34a-5p could inhibit cell autophagy, apoptosis and promote cell proliferation.
Ji, 2021 [114]	Model: OA rat models constructed by cutting the anterior cruciate ligament (the expressions of GAS5 in rat cartilage tissues detected at 4 weeks (early OA) and 12 weeks (late OA) after modeling); then, rat chondrocytes were isolated, cultured and transfected with si-GAS5 to silencing GAS5.	GAS5 Target/signaling pathway: miR-144, mTOR Effects on chondrocytes: After silencing the GAS5, the autophagy ability of OA chondrocytes was increased and the apoptosis rate was decreased; The expression of GAS5 in cartilage tissue of OA rats was higher in late OA than that in early OA.
Wen, 2022 [124]	Model: chondrocytes stimulated with IL-1β to induce chondrocyte injury before adding MSC-Exo for treatment	KLF3-AS1 Target/signaling pathway: YBX1; PI3K/Akt/mTOR Effects on chondrocytes: Repressing autophagy, apoptosis and promoting cell viability
Song, 2022 [119]	Model: knee OA rats constructed in using the modified Hulth method and administered CXTB 35 mg/kg intragastrically for 4 weeks	HOTAIR Target/signaling pathway: p38MAPK Effects on chondrocytes: CXTB improved the morphological structure of the cartilage by enhancing autophagy through downregulation of HOTAIR.
Xu, 2022 [125]	Model: human articular cartilage samples, OA model rats and IL-1β-treated C28/I2 cells	MCM3AP-AS1 Target/signaling pathway: SOX4-activated MCM3AP-AS1; miR-149-5p/Notch1 axis Effects on chondrocytes: Knockdown of MCM3AP-AS1 enhanced autophagy, while alleviated ECM degradation and cartilage injury.
Zhang, 2022 [126]	Model: cartilage tissues harvested from OA patients, OA mouse model established by the destabilization of medial meniscus; their chondrocytes were cocultured with BMSC-EVs overexpressing NEAT1 and NEAT1 was transferred from BMSC-EVs into the chondrocytes.	NEAT1 Target/signaling pathway: miR-122-5p/Sesn2/Nrf2 axis Effects on chondrocytes: Inducing autophagy and the proliferation of chondrocytes but inhibiting the apoptosis; also relieving OA in vivo
Shi, 2022 [127]	Model: OA patients, destabilization of the medial meniscus (DMM) mouse OA model, and IL-1β induced chondrocytes cell; then, intra-articular delivery of lentivirus containing POU3F3 in OA model	POU3F3 Target/signaling pathway: POU3F3/miR-29a-3p/ FOXO3 axis Effects on chondrocytes: Enhancing cell viability, suppressing apoptosis and inflammatory cytokine secretion, rescuing metabolic dysfunction, and restrained autophagy in vitro; alleviating the damage in mouse OA model in vivo

## Data Availability

Not applicable.

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
