# Peer review of "The Role of Autophagy in Osteoarthritic Cartilage"

_biomolecules, 2022, doi:10.3390/biom12101357_

Round 1
Reviewer 1 Report
Kao et al summarized recent studies of autophagy in osteoarthritis, majorly non-coding RNAs. The review provides some valuable information to the readers, but the section structure and writing can be largely improved.
For example,
Section 2 (Autophagy with 2.2 role of autophagy in osteoarthritis, and then 2.3,2.4 different pathways) disconnect from section 3 non-coding RNA in OA. This structure is confusing.
The last paragraph of section 4 can be combined with rapamycin (treatment Cartilage-targeting peptide-modified dual-drug delivery nanoplatform with NIR laser). This is a delivery system for rapamycin.
Pathogenesis of osteoarthritis: some other factors can cause OA, for example, trauma and obesity. Although aging is the main factor other causes may be mentioned.
An outlook may help readers for future direction.
Figures and tables were not referred to in the main text.
Author Response
Reply to reviewer 1 comment
We do appreciate your careful examination of our work, thank you for all the insightful comments!
- Section 2 (Autophagy with 2.2 role of autophagy in osteoarthritis, and then 2.3,2.4 different pathways) disconnect from section 3 non-coding RNA in OA. This structure is confusing.
Answer: Thank you for your valuable comments. We revised the title of 2.3,2.4 (in line 193) and made the article more fluent. Thank you for the suggestion.
- The last paragraph of section 4 can be combined with rapamycin (treatment Cartilage-targeting peptide-modified dual-drug delivery nanoplatform with NIR laser). This is a delivery system for rapamycin.
Answer: Thank you for your valuable comments. We revised the last paragraph (in lines 460-461) and combined it with the paragraph about rapamycin. Thank you for the suggestion, it is helpful for readers to have a better understanding of those treatments.
- Pathogenesis of osteoarthritis: some other factors can cause OA, for example, trauma and obesity. Although aging is the main factor other causes may be mentioned.
Answer: Thank you for your valuable comments. We added more descriptions (in lines 77-79) about risk factors causing OA. Thank you for the suggestion, it is helpful for readers to get a better understanding of OA.
- An outlook may help readers for future direction.
Answer: Thank you for your valuable comments. We added some descriptions about the outlook (in lines 473-476) in the section of conclusions. Thank you for the suggestion, it is helpful for readers to get a better understanding of future direction.
- Figures and tables were not referred to in the main text.
Answer: Thank you for your valuable comments. We rechecked and mentioned about those figures (in lines 138,147,159,203,229,339,354, 423,434,445) and tables (in lines 338-339,353) in the main text. Thank you for the suggestion.

Reviewer 2 Report
Review: “The role of autophagy in osteoarthritis”
My comments are as follows:
The title should be modified reflecting the review. The authors discussed about the role of autophagy in osteoarthritic cartilage.
Lines 55-57 of the pdf: “There is report revealing that OA is the second-most cause, around 7.1%, of musculoskeletal diseases’ burden and the burden of OA considerably raised 63.1% from 1990 to 2007.” References should be provided.
Lines 57-60: Again, references are missing.
Lines 64-66: “Hence, in addition to the ap-64 plication in cancer, autophagy may be a surrogate for OA which numerous research are focusing on [7].” This sentence is not linked with the previous one.
Section 1.2
Lines 68-76: references are absent. There are also other risk factors for OA.
Before discussing about risk factors in Oa, it should be mentioned that OA is a whole joint disease involving all joint tissues (cartilage, meniscus, subchondral bone, infrapatellar fat pad and synovial membrane).
Line 85: references should be added.
Lines 85-95: the authors discuss about apoptosis and authophagy in OA cartilage/chondrocytes. However, it should be mentioned also chondroptosis.
The authors did not report the aim of the review in the text of the manuscript.
Even if this is a narrative review, a brief paragraph about the methods used should be added (i.e. database used and criteria used ect).
Figure 1 seems to be not cited in the text. The caption must be improved describing the figure.
It is not clear to me how did the authors select to mention only some miRNA involved in OA cartilage/chondrocytes and autophagy (section 3.2). The authors should check and improved this part. For example, miRNA-411, miRNA-107 and miRNA-378 regulate autophagy in OA cartilage/chondrocytes.
A table summarizing miRNA involved in OA autophagy should be added.
Section 3.3: Circ-MELK should be added.
Figure 2 should be updated with the miRNA/Circ-RNA not listed. Caption should be improved.
Lines 401-404: could the authors better explain this part?
Section 4: could the authors better introduce this part? What is the idea behind an autophagy treatment for OA? Activating autophagy only in cases of end-stage OA? How did the authors select to report only these substances? There are other substances such as Salvianolic acid B, Isoimperatorin etc.
Lines 99-101, 205-206, 241-243,386-389, 403-404: references are missing.
Abbreviations should be defined at first mention and used consistently throughout the manuscript (i.e. OA).
English needs to be improved. Contracted forms should be not used.
Author Response
Reply to reviewer 2 comment
We do appreciate your careful examination of our work, thank you for all the insightful comments!
- The title should be modified reflecting the review. The authors discussed about the role of autophagy in osteoarthritic cartilage.
Answer: Thank you for your valuable comments. We revised the title and made it more accurately reflecting the review. Thank you for the suggestion.
- Before discussing about risk factors in Oa, it should be mentioned that OA is a whole joint disease involving all joint tissues (cartilage, meniscus, subchondral bone, infrapatellar fat pad and synovial membrane).
Answer: Thank you for your valuable comments. We added more descriptions about OA (in lines 70-72) in this paragraph. Thank you for the suggestion, it produces a better connection and makes the whole paragraph more readable.
- Lines 85-95: the authors discuss about apoptosis and autophagy in OA cartilage/chondrocytes. However, it should be mentioned also chondroptosis.
Answer: Thank you for your valuable comments. We added more descriptions about chondroptosis (in lines 101-105) in the paragraph. Thank you for the suggestion, it produces a more intact concept of chondrocyte death and would be helpful for readers to get a better understanding.
- The authors did not report the aim of the review in the text of the manuscript.
Answer: Thank you for your valuable comments. We added descriptions about the aim of this review (in lines 44-45) in the text briefly. Thank you for the suggestion, it is helpful for readers to understand the purpose of the article.
- Even if this is a narrative review, a brief paragraph about the methods used should be added (i.e. database used and criteria used ect).
Answer: Thank you for your valuable comments. We added descriptions about database and criteria we used (in lines 45-46) in the text briefly. Thank you for the suggestion, it is helpful for readers to get a better understanding of how the article was yielded.
- It is not clear to me how did the authors select to mention only some miRNA involved in OA cartilage/chondrocytes and autophagy (section 3.2). The authors should check and improved this part. For example, miRNA-411, miRNA-107 and miRNA-378 regulate autophagy in OA cartilage/ chondrocytes.
Answer: Thank you for your valuable comments. We added those miRNAs mentioned above (in lines 296-298,300-301) according to your kind suggestion.
- A table summarizing miRNA involved in OA autophagy should be added.
Answer: Thank you for your valuable comments. We added a paragraph (in lines 302-308) to describe that there are still many other miRNAs to be explored for building up the complex autophagic network. Thank you for the suggestion, it is helpful for readers to get a better understanding of how miRNAs were involved in autophagic activity.
- Lines 401-404: could the authors better explain this part?
Answer: Thank you for your valuable comments. We revised the description about treatment of vitamin D (in lines 438-440) in the main text. Thank you for the suggestion, it is helpful for readers to get a better understanding of this paragraph.
- Section 4: could the authors better introduce this part? What is the idea behind an autophagy treatment for OA? Activating autophagy only in cases of end-stage OA? How did the authors select to report only these substances? There are other substances such as Salvianolic acid B, Isoimperatorin etc.
Answer: Thank you for your valuable comments. We chose some examples which is commonly seen and used in our medical environment to show the chance to broaden OA treatment through the mechanism of autophagy. Thank you for the suggestion.
- Lines 55-57 of the pdf: “There is report revealing that OA is the second-most cause, around 7.1%, of musculoskeletal diseases’ burden and the burden of OA considerably raised 63.1% from 1990 to 2007.” References should be provided.
Answer: We added the reference (in line 59) in the main text. Thank you for the suggestion.
- Lines 57-60: Again, references are missing.
Answer: We added the reference (in line 62) in the main text. Thank you for the suggestion.
- Lines 64-66: “Hence, in addition to the ap-64 plication in cancer, autophagy may be a surrogate for OA which numerous research are focusing on [7].” This sentence is not linked with the previous one.
Answer: Thank you for your valuable comments. We added more descriptions (in lines 66-67) in the main text and made the sentence connected to the previous one. Thank you for the suggestion.
- Lines 68-76: references are absent. There are also other risk factors for OA.
Answer: Thank you for your valuable comments. We added references (in lines 74,76,77,79,82) and more description (in lines 77-79) about risk factors for OA in the main text according to your kind suggestion.
- Line 85: references should be added.
Answer: We added the reference in the main text in line 91. Thank you for the suggestion.
- Figure 1 seems to be not cited in the text. The caption must be improved describing the figure.
Answer: Thank you for your valuable comments. We revised the caption of Figure 1 (in lines 245-247) and cited it (in lines 138,147,159, 203,229,423, 434,445) in the main text. Thank you for the suggestion, it is helpful for readers to get a better understanding of the regulation and interaction between the pathways.
- Section 3.3: Circ-MELK should be added.
Answer: Thank you for your valuable comments. We added circ-MELK (in lines 395-402) in section 4.4 according to your kind suggestion.
- Figure 2 should be updated with the miRNA/Circ-RNA not listed. Caption should be improved.
Answer: Thank you for your valuable comments. We updated the miRNA/Circ-RNA not listed in Figure 2 and revised the caption of it (in lines 414-416). Thank you for the suggestion, it produces a more intact introduction of circRNA regulation in autophagy.
- Lines 99-101, 205-206, 241-243,386-389, 403-404: references are missing.
Answer: Thank you for your valuable comments. We added those references (in lines 112,219,258,426,441) in the main text. Thank you for the suggestion, it is helpful for readers to find the material they want more easily.
- Abbreviations should be defined at first mention and used consistently throughout the manuscript (i.e. OA).
Answer: Thank you for your valuable comments. We corrected this error in the main text according to your kind suggestion.
- English needs to be improved. Contracted forms should be not used.
Answer: We corrected this error. We have this manuscript English editing by an native English-speaking scholar. Thank you for the suggestion.

Round 2
Reviewer 1 Report
The authors have addressed most of the prior suggestions.
Reviewer 2 Report
No additional comments.